# Socioeconomic disparities in anthropometric status among primary school children: A potential association with school meals

**Mohammed Abdulrahman Alhassan**[1]*, **Fatima Alhasan**[2]

**1** College of Medicine, Department of Pediatrics, Prince Sattam Bin Abdulaziz University. Al-Kharj, Kingdom of Saudi Arabia, **2** Faculty of Medicine, Department of Physiology, Gezira University. Wad-Madani, Sudan

* mhmdarhafeez@yahoo.com

## Abstract

### Objective

To assess the growth and nutritional status of children in primary schools across different socioeconomic groups in Wad-Madani City, Central Sudan, and map it to World Health Organization (WHO) standards; and to investigate a potential association between school meal intake and nutritional status.

### Methods

This cross-sectional anthropometric study involved a randomly selected sample of 506 children from 10 primary schools in the city. Height and weight were measured following WHO standards and converted into Z-scores for weight-for-age (WAZ), height-for-age (HAZ), and BMI-for-age (BAZ). We compared the mean Z-scores between children in the private and public school sectors, adjusting for ethnicity and other potential predictors. Statistical analyses included multivariate linear regression to assess predictors of growth and nutritional status, alongside group comparisons using appropriate statistical tests.

### Results

Children in public schools had significantly lower BAZ and HAZ levels compared to both WHO standards and private school children. The mean BAZ was -1.0 (SD = 1.23) for public school children and -0.13 (SD = 1.40) for private school children (p = 0.001), with 17.8% (n = 57) of public school children classified as thin (wasted) or severely wasted. The median HAZ was -0.20 (95% CI: -0.34, -0.02) for public school children and 0.19 (95% CI: 0.03, 0.40) for private school children (p < 0.001). Additionally, children in suburban public schools had a significantly lower mean HAZ (-0.46, SD = 11.33) compared to those in urban public schools (p = 0.009). Compared to WHO growth standards, public school children had significantly lower mean WAZ (p < 0.001), HAZ (p = 0.002), and BAZ (p < 0.001). Children who received school meals had significantly higher WAZ (mean difference = 0.619, p = 0.001), HAZ (mean difference = 0.401, p = 0.010), and BAZ (mean difference = 0.588, p = 0.003) across the entire sample. Even within the public-school subgroup,

**Data availability statement:** All relevant data are within the manuscript and its Supporting Information files.

**Funding:** The authors extend their appreciation to Prince Sattam bin Abdulaziz University for funding this research work through the project number (PSAU/2024/03/29191). The funder had no role in study design, data collection and analysis, decision to publish, or preparation of the manuscript.

**Competing interests:** The authors have declared that no competing interests exist.

while statistical significance was not reached, all three parameters—WAZ (mean difference = 0.334, p = 0.074), HAZ (mean difference = 0.262, p = 0.123), and BAZ (mean difference = 0.299, p = 0.132)—remained consistently higher among those who received school meals.

## Conclusion

Public school children exhibit unfavorable growth and nutritional status, which may be attributed to inadequate nutritional and calorie intake. School meals may improve nutritional outcomes. We propose urgent intervention through the provision of nutritionally adequate school meals.

## 1. Introduction

Nutritional status is an important index of children's physical, emotional, and psychological health and well-being. Poor nutritional status in children can lead to a multitude of adverse short and long-term outcomes, including increased susceptibility to and higher rates of death from acute infections in young children [1], impaired cognitive capacities and developmental potential with poor school performance [2], higher risk of nutrition-related chronic conditions in adulthood, and adverse economic implications for individuals and societies [3].

Anthropometric surveys are efficient in assessing the nutritional status and screening for malnutrition in children in a community [4]. Childhood malnutrition in a population or subpopulation is reflected as a deviation of anthropometric growth parameters from population standards. Nutritional assessment of children in the community is pivotal in detecting such deviations and instituting appropriate public health investigations and interventions. The nutritional status of schoolchildren can be used as a proxy for their general health in a community [5].

Anthropometric surveys can be easily conducted by measuring weight, height, and body mass index (BMI) and comparing the results with published growth standards. Acute undernutrition causes fat and muscle wasting (thinness) and can be detected as reduced weight for height or body mass index (BMI) for age. In addition to thinness, long-term undernutrition causes stunting, as indicated by reduced height for age [6]. No official child growth standards exist for Sudanese children. The World Health Organization growth standards are used for the point assessment of growth, nutritional status, and tracking of growth patterns in Sudanese children. The WHO also has a published classification of nutritional status according to growth standard deviation (Z) scores [7].

The prevalence of malnutrition varies across regions. According to the UNICEF-World Health Organization (WHO)-World Bank Joint Child Malnutrition (Global) Estimates 2023 report, 45.0 million children under five were wasted, of which 13.7 million were severely wasted, and 148.1 million were stunted. The highest prevalence of "wasting" was in Sub-Saharan Africa (14.9%), followed by South Asia (7.6%). The highest prevalence of stunting was in South Asia (36.3%), followed by Sub-Saharan Africa (29.9%). More than 340 million children and adolescents aged 5–19 years are overweight or obese [8]. In Sudan, UNICEF-WHO estimated in 2020 that 6.3% of children under the age of 5 were wasted, 4.5% were severely wasted, and 38% were stunted, equivalent to 1.2 million, 0.3 million, and 2.9 million children, respectively [9].

To our knowledge, no previously published study has investigated the prevalence of malnutrition among schoolchildren in Wad-Madani, the largest City in Central Sudan. We have observed that thinness (wasting) and stunting were prevalent among schoolchildren

from particular socioeconomic groups and communities in the City. We hypothesize inadequate nutrient and calorie intake as major contributors to their nutritional status. Although verifying whether dietary inadequacies are the primary cause of abnormal growth requires a dietary survey, an anthropometric survey is still an indispensable first step in portraying the landscape. These observations have not been verified by formally studying and comparing the growth distribution among local schoolchildren in Wad-Madani City.

This anthropometric survey study seeks to assess the growth and nutritional status of primary school children across different socioeconomic groups in Wad-Madani City and map it against WHO standards. In addition, we investigated a potential association between school meal intake and nutritional status. This assessment aims to provide insights into the nutritional status of children in this specific region and is not intended to be generalized to the entire country. It can be used to estimate the prevalence of both undernutrition and overnutrition and explore the potential benefit of providing free or subsidized school meals to children in impoverished communities. We believe these may represent appropriate initial steps in further investigations into the root causes and in designing targeted public health interventions to improve child nutrition in local communities within the city.

## 2. Materials and methods

This anthropometric cross-sectional survey was conducted from September 2021 to March 2022 among children from 10 primary schools in Wad-Madani City. Wad-Madani is the second largest city in entire Sudan, with a population of approximately 404,000 in 2023 [9]. Pupils from two private sector schools and eight public schools (six central/urban schools and two peripheral or suburban schools) were recruited. With a population of around 90000 primary school children in Wad-Madani, a hypothesized prevalence of malnutrition of 15% [10], a confidence limit of 5% and, a cluster effect of 2, we calculated a minimum required sample size of 392 for a 95% confidence level. A simple random sample of children aged 5–10 years was selected from each school using a list of enrollees in grades matching the eligible age groups. A simple manual selection method was employed, where names of all listed students were drawn randomly by the investigator. Those selected were checked for eligibility, and if found ineligible, replacements were drawn using the same random selection process. Selection within each school was not proportionate to school enrollment size; instead, an equal opportunity approach was applied to ensure feasibility in recruitment across schools. The school sector was used as a proxy for socioeconomic class. This decision was based on local challenges in Sudan, where household income is often unstable, culturally sensitive to disclose, and subject to rapid currency fluctuations, making direct classification impractical. Standard socioeconomic measures have also been difficult to apply in this context. While not absolute, school sector remains a practical and widely used proxy in similar research settings. Inclusion criteria were healthy primary school children aged 5–10 years, while children with chronic health conditions or uncertain birth dates were excluded.

A questionnaire was sent to the participants' parents/guardians to collect sociodemographic information, including date of birth and ethnicity of origin. Height and weight were measured by the investigator FA, who is a medical doctor. Standard techniques and scales were used for measurements. Height measurement involved the child standing against a wall with the child's head in the Frankfurt plane, heels together, and back straight against the wall. Using a calibrated hard tape measure, heights were measured using headboard demarcations in the wall. A calibrated, standing weight measuring scale was used for weight measurement. The weight-for-age, height-for-age, and BMI-for-age data were transformed into standard deviation (Z) scores (SDS) based on WHO growth standards. Online software by the WHO was used to calculate the

Z-score for each child (ANTHROPLUS Software for Assessing Growth of the World's Children and Adolescents. Geneva: WHO, https://www.who.int/tools/growth-reference-data-for-5to19-years/application-tools). The primary outcome variables were weight-for-age (WAZ), height-for-age (HAZ), and BMI-for-age (BAZ) Z-scores. We specifically investigated a potential association with school meal intake. The World Health Organization (WHO) cutoff points for the classification of nutritional status were used (see below under the results section). We compared the mean WAZ, HAZ, and BAZ scores of the children in the two school sectors (private versus public), adjusting for ethnicity and other potential predictors.

In this study, permissions were obtained from the local Ministry of Health and Ministry of Education in Wad Madani City and from each school administration. Informed written consent was obtained from all subjects involved in the study. Prior to data collection, a pen-and-paper questionnaire was sent to the children's parents or guardians, seeking their informed written consent for their child's participation. Only children whose parents returned the informed written consent were included in the study. Additional information regarding the ethical, cultural, and scientific considerations specific to inclusivity in global research is included in the Supporting Information (S1 File).

The study received ethical approval from the Health Sector Ethical Review Committee, University of Gezira (No: 6/2021). The study did not involve interventions to participants or the collection of human specimens, and data was sufficiently anonymized to ensure privacy and confidentiality. While no names or personal addresses were collected, date of birth was used to calculate age. The study adhered to the Declaration of Helsinki for conducting research involving human participants.

Data were analyzed using STATA software statistical package (version 17.0). Descriptive statistics were summarized as frequencies and percentages for nominal data and medians and interquartile ranges or means and standard deviations for continuous data. Normality was assumed for the anthropometric data and visually verified using histograms. A one-sample t-test was used to compare the means of WAZ, HAZ, and BAZ in our entire sample and the school sector samples with the WHO population mean (zero Z-score). Two-sample t-tests were used to compare the WAZ, HAZ, and BAZ between the two school sectors. One-way ANOVA was used to compare outcome measures across the levels of other categorical variables. Univariate analyses were performed to screen for potential predictors. Predictors with P-values of more than 0.25 were included in the multiple regression model construction. A multivariate linear regression model containing the primary predictor variables (school sector as a proxy for socioeconomic status) and other candidate variables was constructed via backward elimination to control for potential confounders and evaluate other potential predictors of nutritional status. School meal intake was similarly investigated as a potential predictor controlling for school sector categories. A P-value of 0.05 was considered statistically significant, and 95% confidence intervals (CI) are quoted where appropriate.

## 3. Results

### 3.1. Sociodemographic characteristics

Out of 1000 consent forms sent to guardians of primary school children, 506 returned the form and were recruited in the study. Participants were categorized according to the school sector into public (320 children, 63.2%) and private (186 children, 36.8%). The former was subdivided into public urban schools (244 children) and public suburban schools (76 children) for subgroup analyses. Of the children included, 259 (51.2%) were female and 247 (48.8%) were male. The age categories and other sociodemographic characteristics of the participants are presented in Table 1.

**Table 1. Sociodemographic characteristics of children by school sector[1].**

| Variable | School sector | | Total | P-value |
|---|---|---|---|---|
| | Public 320 (63.2%) | Private 186 (36.8%) | 506 | |
| **Age category (years)** | | | | < 0.001 |
| 5-6 | 7 (2.2%) | 11 (5.9%) | 18 (3.6%) | |
| 6-7 | 67 (20.9%) | 45 (24.2%) | 112 (22.1%) | |
| 7-8 | 95 (29.7) | 43 (23.1%) | 138 (27.3%) | |
| 8-9 | 71 (22.2%) | 62 (33.3%) | 133 (26.3%) | |
| 9-10 | 80 (25.0%) | 25 (13.4%) | 105 (20.8%) | |
| **Gender** | | | | 0.438 |
| Female | 168 (52.5%) | 91 (48.9%) | 259 (51.2%) | |
| Male | 152 (47.5%) | 95 (51.1%) | 247 (48.8%) | |
| **Ethnicity** | | | | < 0.001 |
| Arab | 162 (50.6%) | 137 (73.7%) | 299 (59.1%) | |
| African | 83 (25.9%) | 7 (3.8%) | 90 (17.8%) | |
| Nubian | 16 (5.0%) | 9 (4.9%) | 25 (4.9%) | |
| Bija | 4 (1.3%) | 9 (4.8%) | 13 (2.6%) | |
| Other/mixed | 55 (17.2%) | 24 (12.9%) | 79 (15.6%) | |
| **Household members** | | | | < 0.001 |
| <5 | 72 (22.9%) | 91 (50.8%) | 163 (33.0%) | |
| 6-10 | 184 (58.4%) | 76 (42.5%) | 260 (52.6%) | |
| >10 | 59 (18.7%) | 12 (6.7%) | 71 (14.4%) | |
| Total | 315 | 179 | 494 | |
| **Family principal income earner** | | | | 0.006 |
| Father | 217 (69.6%) | 144 (82.3%) | 361 (74.1%) | |
| Mother | 9 (2.9%) | 6 (3.4%) | 15 (3.1%) | |
| Both | 72 (23.1%) | 23 (13.1%) | 95 (19.5%) | |
| Others | 14 (4.5%) | 2 (1.1%) | 16 (3.3%) | |
| Total | 312 | 175 | 487 | |
| **Parental status** | | | | 0.083 |
| Married | 288 (93.2%) | 165 (93.8%) | 453 (93.4%) | |
| Widowed | 10 (3.2%) | 1 (0.6%) | 11 (2.3%) | |
| Divorced | 11 (3.6%) | 9 (5.1%) | 20 (4.1%) | |
| Both died | 0 (0.0%) | 1 (0.6%) | 1 (0.2%) | |
| Total | 309 | 176 | 485 | |
| **Mother education** | | | | < 0.001 |
| Illiterate | 26 (8.6%) | 1 (0.6%) | 27 (5.6%) | |
| Primary | 61 (20.3%) | 5 (2.8%) | 66 (13.7%) | |
| Secondary | 71 (23.6%) | 37 (20.6%) | 108 (22.5%) | |
| University | 123 (40.9%) | 104 (57.8%) | 227 (47.2%) | |
| Postgrad | 20 (6.6%) | 33 (18.3%) | 53 (11.0%) | |
| Total | 301 | 180 | 481 | |
| **Working mother** | | | | 0.168 |
| No | 188 (61.2%) | 118 (67.8%) | 306 (63.6%) | |
| Yes | 119 (38.8%) | 56 (32.2%) | 175 (36.4%) | |
| Total | 307 | 174 | 481 | |
| **Duration of breastfeeding of child** | | | | 0.984 |
| None | 7 (2.9%) | 3 (1.7%) | 10 (2.4%) | |

*(Continued)*

**Table 1.** (Continued)

| Variable | School sector | | Total | P-value |
|---|---|---|---|---|
| | **Public**<br>**320 (63.2%)** | **Private**<br>**186 (36.8%)** | **506** | |
| 1-6 months | 12 (5.0%) | 11 (6.2%) | 23 (5.5%) | |
| 7-12 months | 24 (10.0%) | 18 (10.1%) | 42 (10.0%) | |
| 13-18 months | 76 (31.5%) | 60 (33.7%) | 136 (32.5%) | |
| 19-24 months | 122 (50.6%) | 86 (48.3%) | 208 (49.6%) | |
| Total | 241 | 178 | 419 | |
| **Monthly income category** | | | | < 0.001 |
| Have to debt (must borrow to cover needs) | 64 (20.8%) | 15 (8.5%) | 79 (16.3%) | |
| Insufficient (struggling but no debt) | 123 (39.9%) | 21 (11.9%) | 144 (29.7%) | |
| Sufficient | 109 (35.4%) | 102 (57.6%) | 211 (43.5%) | |
| Saving | 12 (3.9%) | 39 (22.0%) | 51 (10.5%) | |
| Total | 308 | 177 | 485 | |
| **School meal intake** | | | | < 0.001 |
| No | 44 (14.0%) | 5 (2.8%) | 49 (9.9%) | |
| Yes | 270 (86.0%) | 176 (97.2%) | 446 (90.1%) | |
| Total | 314 | 181 | 495 | |

[1]Chi-squared test was used unless otherwise specified. Fisher's exact test was applied where expected cell counts were < 5.

### 3.2. BAZ by school sector

The median BAZ was -0.86 (95%CI: -0.97, -0.69). There was a significant difference in the mean BAZ between public (-1.0, SD = 1.23) and private school children (-0.13, SD = 1.40, p < 0.001) (Table 2). Children in suburban public schools had a significantly lower mean BAZ than those in urban schools (two-tailed p = 0.017) (Fig 1).

### 3.3. BAZ compared to WHO standards

The mean BAZ (-0.68, SD = 1.36) was significantly lower than that of the WHO population (p < 0.001). However, the mean score for private schools was not significantly lower than the WHO mean score (p = 0.211). Children in public schools had a significantly lower mean BAZ than those in WHO schools (p < 0.001). In addition, over 2/3rd of the children in public schools had BAZ < zero (Table 3 and Figs 1 and 2).

### 3.4. Nutritional status according to BAZ

Of the entire sample, 13.5% were either 'wasted' (thin) or severely wasted, and 5.3% were obese. These percentages were significantly higher for thinness and lower for obesity in public school children than in private schools (17.8% vs. 5.9%; 3.1% vs. 9.1%, respectively; P < 0.001) (Table 2). This significant difference was observed after adjusting for ethnicity as a potential confounder (p < 0.007). 12.4% of all sampled children were either overweight or obese. This percentage increased to 21.5% in private school children, where 12.4% were overweight and 9.1% were obese (Table 2).

### 3.5. HAZ by school sector

The median HAZ was -0.20 (95%CI: -0.34, -0.02) for public school children and 0.19 (95%CI: 0.03, 0.40) for private schools. There was a significant difference in the mean HAZ between

**Table 2. Mean growth Z-scores and anthropometric status of children.**

| Measure | School sector | | Combined (n = 506) | P-value |
|---|---|---|---|---|
| | Public (n = 320) | Private (n = 186) | | |
| **BAZ** | | | | < 0.001* |
| mean (SD), | -1.00 (1.23), | -0.13 (1.40), | -0.68 (1.36), | |
| median (95% CI) | -1.12 (-1.31, -0.99) | -0.41 (-0.56, -0.13) | -0.86 (-0.97, -0.69) | |
| **Nutritional status according to BAZ** | | | | < 0.001 |
| Severe wasting (severe thinness) (<-3) | 8 (2.5%) | 3 (1.6%) | 11 (2.2%) | |
| Wasting (thinness) (<-2 to -3) | 49 (15.3%) | 8 (4.3%) | 57 (11.3%) | |
| Normal (-2 to 1) | 240 (75.0%) | 135 (72.6%) | 375 (74.1%) | |
| Overweight (> 1 to 2) | 13 (4.1%) | 23 (12.4%) | 36 (7.1%) | |
| Obese (> 2) | 10 (3.1%) | 17 (9.1%) | 27 (5.3%) | |
| **HAZ** | | | | < 0.001 $ |
| mean (SD), | -0.18 (1.07), | 0.15 (1.03), | -0.06 (1.07), | |
| median (95% CI) | -0.20 (-0.34, -0.02) | 0.19 (0.03, 0.40) | -0.01 (-0.14, 0.12) | |
| **Stature classification** | | | | 0.735 # |
| Severe stunting (<-3) | 2 (0.6%) | 1 (0.5%) | 3 (0.6%) | |
| Stunting (<-2) | 8 (2.5%) | 3 (1.6%) | 11 (2.2%) | |
| Normal (-2 to 2) | 305 (95.31 | 176 (94.62 | 481 (95.1%) | |
| Tall (> 2) | 4 (1.3%) | 5 (2.7%) | 9 (1.8%) | |
| Very tall (> 3) | 1 (0.3%) | 1 (0.5%) | 2 (0.4%) | |
| **Concurrent stunting and wasting (decompensated chronic undernutrition) (BAZ between -2 and 2 and HAZ <-2)** + | 10 (3.13%) | 3 (1.61%) | 13 (2.57%) | 0.300 |
| **WAZ,** | | | | < 0.001 & |
| mean (SD), | -0.76 (1.15), | 0.02 (1.26), | -0.48 (1.25), | |
| median (95% CI) | -0.86 (-0.96, -0.66) | -0.05 (-0.19, 0.18) | -0.53 (-0.66, -0.44) | |
| **Nutritional status according to WAZ** | | | | 0.008 |
| Severely underweight (<-3) | 5 (1.6%) | 1 (0.5%) | 6 (1.2%) | |
| Underweight (<-2 to -3) | 32 (10.0%) | 9 (4.8%) | 41 (8.1%) | |
| Normal (-2 to 2) | 276 (86.3%) | 162 (87.1%) | 438 (86.6) | |
| > 2 – 3 SD (Z) | 6 (1.9%) | 10 (5.4%) | 16 (3.2%) | |
| > 3 SD (Z) | 1 (0.3%) | 4 (2.2%) | 5 (1.0%) | |

BAZ, BMI for age z-score; HAZ, height for age z-score; WAZ, weight for age z-score.

*This significant difference in BAZ remained when adjusting for ethnicity (a nonmodifiable potential predictor) using linear regression (P-value < 0.001).

$This significant difference in HAZ remained when adjusting for ethnicity (a nonmodifiable potential predictor) using linear regression (P-value = 0.007).

#Categorizing a continuous variable can result in a loss of information. When using the height-for-age z-score as a continuous variable, the difference was statistically significant, as shown in the same table.

+Adapted from Table 1 in the References. [6].

&This significant difference in WAZ holds when adjusting for ethnicity (a non-modifiable potential predictor) using linear regression (P-value < 0.001).

public (-0.18, SD = 1.07) and private school children (0.15, SD = 1.03) (p < 0.001) (Table 3 and Fig 1). In addition, suburban public-school children had a significantly lower mean HAZ (-0.46, SD = 1.13) than those in urban public schools (-0.10, SD = 1.04), with a two-tailed p = 0.009 (Fig 1).

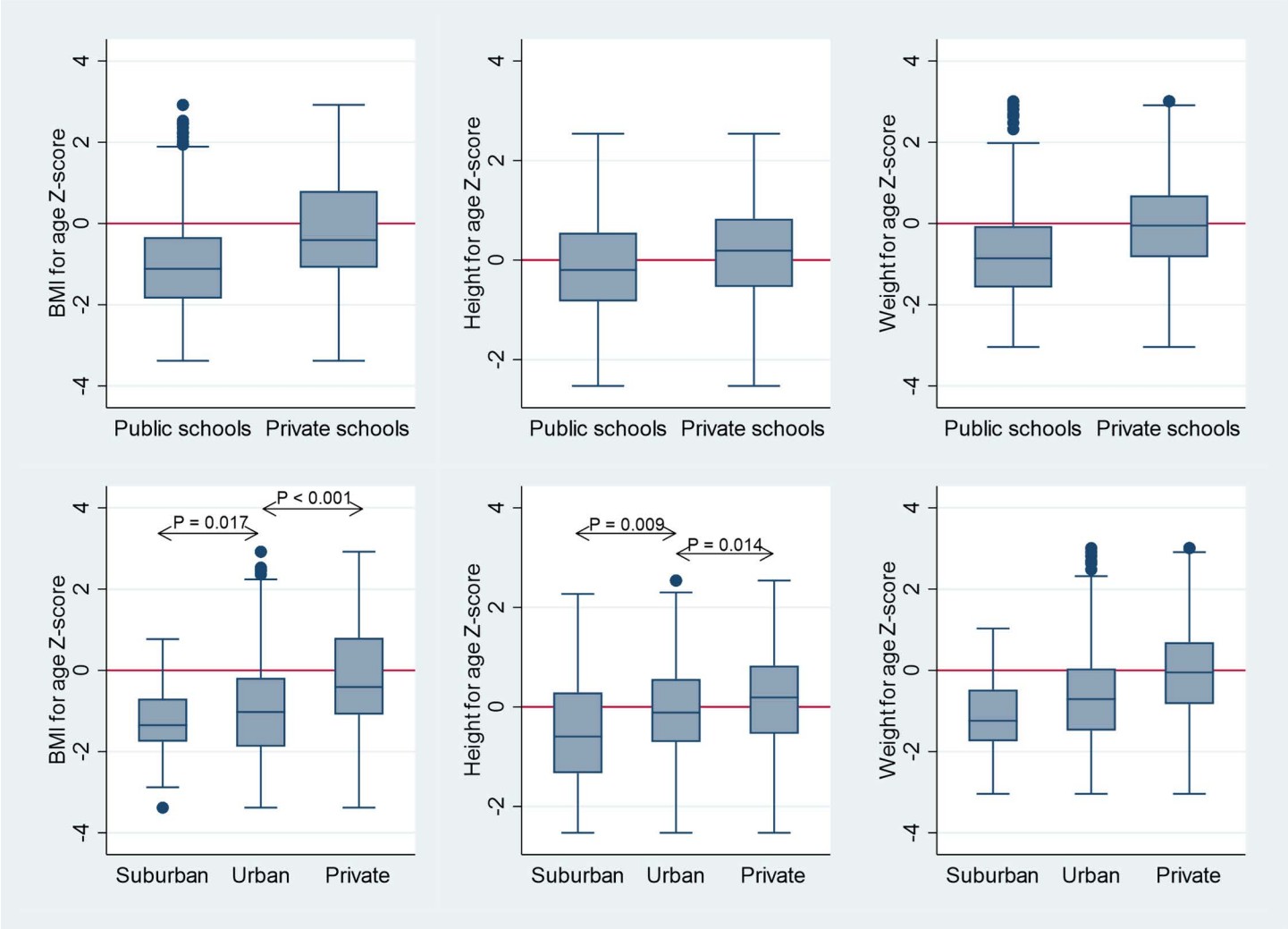

**Fig 1. Box plot for BMI, height, and weight Z-scores by school sector (upper row) and with public schools sub-grouped into urban and suburban (lower row).** For the purpose of these figures, 1 percent at each tail of the distribution has been trimmed. This has only removed outliers from figures, with no effect on boxes or whiskers. Trimming has not been applied during data analysis.

### 3.6. HAZ compared to WHO standards

While the mean HAZ (-0.06, SD = 1.07) for the entire sample was not significantly different from the WHO mean (p = 0.201), the mean HAZ was significantly lower (p = 0.002) for public schools and significantly higher (p = 0.046) for private schools than the WHO mean (Table 3 and Fig 1).

### 3.7. Stature classification

Table 2 summarizes the stature and nutritional status of the participants according to the HAZ and WAZ.

### 3.8. Multivariable linear regression

The details of the model building are discussed in the Methods section. Table 4 presents the multiple linear regression analysis results for the BAZ, HAZ, and WAZ. Affiliation with the

**Table 3. Comparison of anthropometric measurements to WHO standards.**

| Statistic | Observations | Mean | Standard deviation | P-value (2-tailed) | 95% CI |
|---|---|---|---|---|---|
| **BAZ combined** | 506 | -0.68 | 1.36 | **< 0.001** | -0.80, -0.56 |
| **BAZ public schools** | 320 | -1.00 | 1.24 | **< 0.001** | -1.14, -0.87 |
| **BAZ private schools** | 186 | -0.13 | 1.40 | 0.211 | -0.33, 0.07 |
| **BAZ suburban public schools** | 76 | -1.30 | 0.76 | **< 0.001** | -1.47, -1.12 |
| **BAZ urban public schools** | 244 | -0.91 | 1.34 | **< 0.001** | -1.08, -0.74 |
| **HAZ combined** | 506 | -0.06 | 1.07 | 0.201 | -0.15, 0.03 |
| **HAZ public schools** | 320 | -0.18 | 1.07 | **0.002** | -0.30, -0.07 |
| **HAZ private schools** | 186 | 0.15 | 1.03 | **0.046** | 0.00, 0.30 |
| **HAZ suburban public schools** | 76 | -0.46 | 1.13 | **< 0.001** | -0.72, -0.21 |
| **HAZ urban public schools** | 244 | -0.10 | 1.04 | 0.149 | -0.23, 0.03 |

BAZ: BMI for age z-score; HAZ: height-for-age z-score.

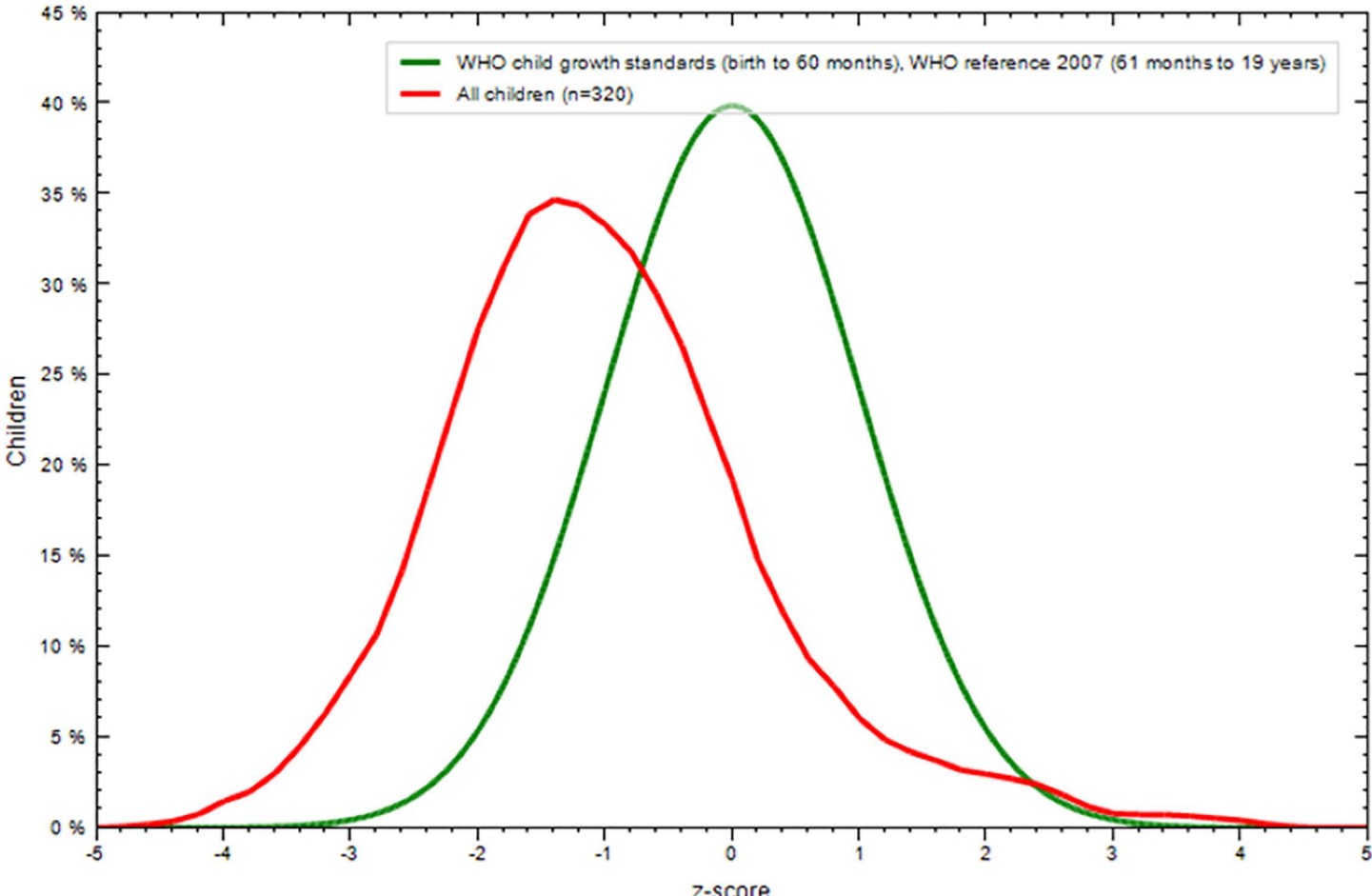

**Fig 2. Histogram of BMI Z-score against WHO standards for public school children.** (Figure derived from ANTHROPLUS Software for Assessing Growth of the World's Children and Adolescents. Geneva: WHO, https://www.who.int/tools/growth-reference-data-for-5to19-years/application-tools).

**Table 4. Multivariable linear regression analysis 1.**

|  | Coefficient | P-value | 95% CI | Prob > F | Adjusted R² |
|---|---|---|---|---|---|
| **BAZ** |  |  |  | < 0.001 | 0.12 |
| School sector | 0.71 | **< 0.001** | 0.44 - 0.98 |  |  |
| Mother education | 0.18 | **0.005** | 0.05 - 0.30 |  |  |
| Income category | 0.10 | 0.160 | -0.04 - 0.25 |  |  |
| Ethnicity | 0.10 | **0.010** | 0.02 - 0.18 |  |  |
| **HAZ** |  |  |  | < 0.001 | 0.03 |
| School sector | 0.24 | **0.022** | 0.04 - 0.45 |  |  |
| Age | -0.07 | 0.099 | -0.15 - 0.01 |  |  |
| Mother education | 0.07 | 0.190 | -0.03 - 0.17 |  |  |
| School meal availability/intake | 0.27 | 0.126 | -0.07 - 0.60 |  |  |
| **WAZ** |  |  |  | < 0.001 | 0.11 |
| School sector | 0.65 | **< 0.001** | 0.39 - 0.90 |  |  |
| Mother education | 0.17 | **0.005** | 0.05 - 0.28 |  |  |
| Income category | 0.08 | 0.267 | -0.06 - 0.21 |  |  |
| Ethnicity | 0.08 | **0.025** | 0.01 - 0.16 |  |  |

BAZ, BMI for age z-score; HAZ, height for age z-score; WAZ, weight for age z-score.

1 Only variables with P-values of > 0.25 on univariate analyses were included in the multiple regression model. The correlated covariates were not included in the model. Backward elimination was then employed to yield the model with the best adjusted R-squared.

school sector was a constant significant predictor, even after adjusting for other important potential predictors, namely, mother's education and ethnicity.

### 3.9. School meal

Analysis of anthropometric measurements revealed significant associations between school meal intake and nutritional status among schoolchildren. Across the entire sample, children who received school meals had significantly higher WAZ (mean difference = 0.619, p = 0.001), HAZ (mean difference = 0.401, p = 0.010), and BAZ (mean difference = 0.588, p = 0.003). Even within the public-school subgroup, while statistical significance was not reached, all three parameters—WAZ (mean difference = 0.334, p = 0.074), HAZ (mean difference = 0.262, p = 0.123), and BAZ (mean difference = 0.299, p = 0.132)—remained consistently higher among those who received school meals (Fig 3).

## 4. Discussion

Arguably, the most remarkable finding in this study was the significantly lower overall growth and nutritional status statistics of public-school children compared to both WHO standards and private school children. On average, children in public schools were significantly thinner and shorter than those in private schools. Furthermore, the children in suburban schools were significantly thinner and shorter than those in urban public schools. These disparities persisted even after adjusting for other non-modifiable factors such as ethnicity and age. A comparison with WHO standards verified that these disparities were not due to overnutrition in private school children; rather, children from public schools were particularly thinner and shorter than the WHO averages. Notably, children in private schools had significantly higher mean heights than the WHO standards. Another remarkable finding is that more than 17% of children in public schools are wasted or severely wasted. This prevalence is comparable with other previous studies from other areas of Sudan (8.26% [11] and

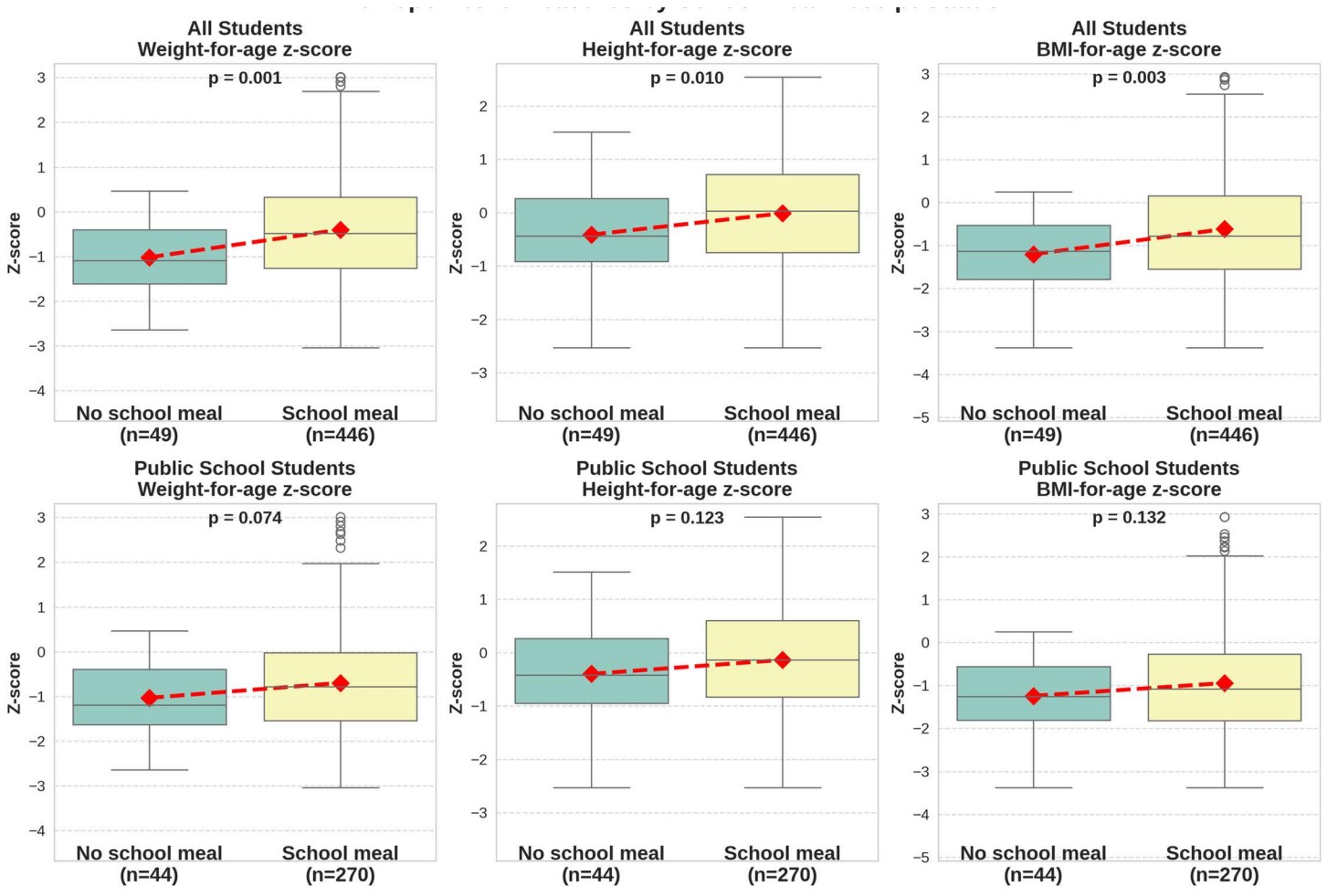

**Fig 3. Effect of school meal intake on WAZ, HAZ, and BAZ in the entire sample (above) and public-school children (below).** Dashed lines connect the means (red diamonds).

23.1% [12]) and the world (10% [13]). The prevalence of stunting or severe stunting (3.1%) in public school children in this study, however, is lower than all reviewed previous studies (7.1% [12], 8% [13], 14.3% [14]), including a meta-analysis in Sub-Saharan Africa, where 40% of boys and 36% of girls were reported as stunted [15]. We could not explain why the average HAZ was lower than the WHO standards while the prevalence of stunting did not sound alarming in public school children. Another remarkable finding was the comparatively high prevalence of overweight and obesity among children in private schools. Approximately one in five children was either obese or overweight. However, this is not unexpected, given that socioeconomic status and obesity appear to be positively correlated in similar populations in low- and middle-income countries [16]. Children from affluent families typically had more access to processed foods and sugary drinks [17]. This is usually coupled with insufficient community awareness of the implications of childhood obesity. These children had less access to physical activity because they lived in urban areas and experienced prohibitive outdoor heat on most days of the year [18].

Affiliation with private, central, or peripheral public schools reflects the socioeconomic background of the children. We believe that these findings are not attributable to ethnic

differences, as significant disparities persisted after adjusting for the non-modifiable factors of ethnicity and age in the statistical analysis. This suggests the presence of other unstudied factors that are important potential predictors for future research. We propose that these shortcomings in the growth and nutritional status of public-school children are directly related to inadequate nutrition and calorie intake. Owing to a lack of resources, we could not verify this through a dietary survey. The effects of other contributing predictors cannot be ignored, as growth and nutritional status are complex interplays between several factors. Poor sanitation, leading to increased incidences of diarrheal disease, parasitic infections, and adverse emotional and psychosocial circumstances, merits particular attention and further investigation. The root causes of these factors are likely poverty and poor parental education. In the multiple regression model used in this study, maternal education was shown to be a constant potential predictor of both BAZ and HAZ, and income was a predictor of BAZ. The low R-squared value in our multiple regression model suggests that there are other significant factors influencing the outcomes that were not included in this study. Future research should aim to investigate and identify these factors.

Therefore, while these results suggest that inadequate nutritional intake is a key factor, they do not establish causality. Findings from our study suggest that caregiver-provided school meals are associated with better nutritional outcomes overall, but this association is less pronounced among public school children. This may be due to differences in baseline nutritional status, socioeconomic factors, and environmental contexts. Despite the differences being statistically insignificant within the public-school subgroup, a potential significant association at the population level should not be dismissed, particularly given the consistent higher values across all three parameters (BAZ, HAZ, and WAZ) for children who had school meals. The small sample size in each subgroup likely reduced the study's power, contributing to the lack of statistical significance. To confirm the effectiveness of providing proper meals at school in reducing wasting and stunting in our context, further studies employing larger samples and intervention, or randomized controlled trial designs are required. However, the effectiveness of school meal programs in improving nutritional status and educational outcomes has been reported. School meal programs to address undernutrition in food-insecure areas have been demonstrated to be beneficial for educational outcomes [19], health, and nutritional status [20]. We propose providing free nutritionally adequate school meals as a plausible urgent intervention. No free or subsidized school meals are currently provided in schools in Sudan. Partnerships between local governments and concerned local and global non-governmental organizations are pivotal for planning and implementing such programs [21]. The application of educational strategies through community campaigns to raise awareness of child nutrition and malnutrition and promote healthy and clean food intake is potentially applicable and effective [22]. However, addressing the root causes of poverty and poor education relies on strategic planning and is outside the scope of this study.

## 5. Conclusions

Public school children exhibit unfavorable growth and nutritional status, which may be attributed to inadequate nutritional and calorie intake. School meals may improve nutritional outcomes. We propose urgent intervention through the provision of nutritionally adequate school meals.

## Supporting information

**S1 File. Inclusivity in global research checklist.**
(DOCX)

**S2 File. Dataset.**
(XLSX)

## Author contributions

**Conceptualization:** Mohammed Abdulrahman Alhassan, Fatima Alhasan.

**Data curation:** Mohammed Abdulrahman Alhassan.

**Formal analysis:** Mohammed Abdulrahman Alhassan.

**Funding acquisition:** Mohammed Abdulrahman Alhassan.

**Investigation:** Fatima Alhasan.

**Methodology:** Mohammed Abdulrahman Alhassan.

**Project administration:** Fatima Alhasan.

**Supervision:** Mohammed Abdulrahman Alhassan.

**Visualization:** Mohammed Abdulrahman Alhassan.

**Writing – original draft:** Mohammed Abdulrahman Alhassan.

**Writing – review & editing:** Fatima Alhasan.

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
