## [Decision Letter · Decision Letter 0]

30 Jan 2025

PONE-D-24-52592Socioeconomic Disparities in Growth and Nutritional Status Among Primary School Children in Central Sudan: A Potential Association with School MealsPLOS ONE

Dear Dr. Alhassan,

Thank you for submitting your manuscript to PLOS ONE. After careful consideration, we feel that it has merit but does not fully meet PLOS ONE’s publication criteria as it currently stands. Therefore, we invite you to submit a revised version of the manuscript that addresses the points raised during the review process.

**ACADEMIC EDITOR: **

Dear Dr Mohammed,

This is an important topic that you have discussed in your manuscript.

Please go through the major comments provided by the reviewers and answer them one by one.

We look forward to receiving your revised manuscript.

Kind regards,

Bilal Ahmad Rahimi, M.D., D.T.M.&H., M.C.T.P., Ph.D

Academic Editor

PLOS ONE

Journal requirements: When submitting your revision, we need you to address these additional requirements. 1. Please ensure that your manuscript meets PLOS ONE's style requirements, including those for file naming. The PLOS ONE style templates can be found at https://journals.plos.org/plosone/s/file?id=wjVg/PLOSOne_formatting_sample_main_body.pdf and https://journals.plos.org/plosone/s/file?id=ba62/PLOSOne_formatting_sample_title_authors_affiliations.pdf. 2. Please include a complete copy of PLOS’ questionnaire on inclusivity in global research in your revised manuscript. Our policy for research in this area aims to improve transparency in the reporting of research performed outside of researchers’ own country or community. The policy applies to researchers who have travelled to a different country to conduct research, research with Indigenous populations or their lands, and research on cultural artefacts. The questionnaire can also be requested at the journal’s discretion for any other submissions, even if these conditions are not met. Please find more information on the policy and a link to download a blank copy of the questionnaire here: https://journals.plos.org/plosone/s/best-practices-in-research-reporting. Please upload a completed version of your questionnaire as Supporting Information when you resubmit your manuscript. 3. Please provide additional details regarding participant consent. In the ethics statement in the Methods and online submission information, please ensure that you have specified (1) whether consent was informed and (2) what type you obtained (for instance, written or verbal, and if verbal, how it was documented and witnessed) 4. Thank you for stating the following financial disclosure:  [The authors extend their appreciation to Prince Sattam bin Abdulaziz University for funding this research work through the project number (PSAU/2024/03/29191).].  Please state what role the funders took in the study.  If the funders had no role, please state: ""The funders had no role in study design, data collection and analysis, decision to publish, or preparation of the manuscript."" If this statement is not correct you must amend it as needed. Please include this amended Role of Funder statement in your cover letter; we will change the online submission form on your behalf. 5. Please include a caption for figure 1, 2 and 3.  6. Please include captions for your Supporting Information files at the end of your manuscript, and update any in-text citations to match accordingly. Please see our Supporting Information guidelines for more information: http://journals.plos.org/plosone/s/supporting-information. 

Reviewers' comments:

Reviewer's Responses to Questions

**Comments to the Author**

1. Is the manuscript technically sound, and do the data support the conclusions?

Reviewer #1: Yes

Reviewer #2: Yes

2. Has the statistical analysis been performed appropriately and rigorously? 

Reviewer #1: Yes

Reviewer #2: Yes

3. Have the authors made all data underlying the findings in their manuscript fully available?

Reviewer #1: Yes

Reviewer #2: Yes

4. Is the manuscript presented in an intelligible fashion and written in standard English?

Reviewer #1: Yes

Reviewer #2: Yes

5. Review Comments to the Author

Reviewer #1: The authors have not categorically pointed out the inculsion and exclusion crieteria for the subject sample. Sample size (506) does not represent the socioeconomic disparities associated with growth and nutritional status of the children at national level hence a larger study is needed to depict if the same situation prevail in the country. There have been spelling and grammar mistakes in the text which need to be addressed. The verbatin while interpreting data has to be avoided as it creats a bit confusion for the readers implying the article should have more clarity.

Reviewer #2: REVIEWER’S COMMENTS

I commend the authors for the efforts put into this study on socioeconomic disparities in growth and nutritional status among primary school children in Central Sudan: A potential association with school meals. The shows potentials for interventions to promote child growth. The following comments may help to improve the manuscript for publication.

Title: Growth should be expunged in the title as direct growth assessment was not conducted. The growth indices assessed depict nutritional status.

Abstract

The method section of the abstract is not detailed enough to show how the objective was achieved. Was comparison of the Z scores between private and public schools the only assessment conducted? The study design, sample size and sampling techniques used are missing. Which data collection methods were employed? How were the data analysed? The title showed that the authors also assessed growth. How was this conducted?

The results are clear but results on growth is missing. It is either removed in the title or the missing part included. By growth, I expected to see baseline and endline values and mean differences that shows increase or decrease in growth parameters (weight, height, BMI). When these parameters are related to age or any other parameter, it becomes an index of nutritional status.

Growth used in the conclusion is inappropriate because growth was not assessed. The authors assessed nutritional status; the growth indicators are not direct assessment of growth pattern.

The key words should also be reviewed in line with this statement.

Introduction

weight for height or body mass index (BMI) for age: these are indices and should be written as compound words.

Materials and methods

1. We estimated a sample size of 392: This statement is inappropriate. The authors estimated the sample size instead of calculating it based on a prevalence of malnutrition of 15%, a confidence limit of 5%, a cluster effect of 2 and a 95% confidence level. Is there any explanation for this? The authors estimated a sample size of 392 but distributed 1000 consent forms. What was the essence of estimating (uncalculated) the sample size. It appears that selection of the pupils for the study was dependent on return of consent forms. It is also not clear how they arrived at the 1000 pupils given the forms. How many children were selected from each of the areas (public/private) schools and on what basis. Same applies to each of the schools.

2. A random sample of healthy children aged 5 and 10 years was selected from each school: How did the authors establish that the children were healthy? There are different types of random sampling. The authors should describe the ones used. Sampling technique is an important aspect of research. The authors should detail step by step how the sample was selected.

3. The school sector was used as a proxy for socioeconomic class: How could this be? Was it extremely difficult to elicit and classify the socioeconomic status of the parents/guardians? Some middle and low socio-economic class parents who appreciate the benefits of good education strive against all odds to offer their children good education.

4. Prior to data collection, a pen-and-paper questionnaire was sent to the children's parents or guardians, seeking their informed consent for their child's participation: How did the authors handle rejection because it is obvious they did not meet the parents to explain.

Results

1. The authors estimated a sample size of 392 but distributed 1000 consent forms to parents/guardians of the primary school children. Out of which 506 were returned recruited in the study. What was the reason for sending out 1000 consent forms? It implies that selection of the pupils for the study was dependent on return of consent forms. It is also not clear how they arrived at the 1000 pupils given the forms.

2. Table 1: Please change Character to variables.

3. The total after each variable should be expunged since there is total beneath the headings.

4. The m in the table should be written in full or explained as a footnote.

5. Monthly income category: The options under this variable is not clear. Example, Have to debt and saving do not show monthly income.

6. School meal availability/intake: This is not clear. Availability and intake (consumption) are not the same.

7. Chi-squared or Fisher’s exact: Which one did the authors use?

8. Nutritional status and anthropometric measurements by school sector: What is the difference between nutritional status and anthropometric measurements? There are no weight and height measurements values in the table.

9. Figure 3: Pupils from public schools were 320. Out of this, 270 consume school meals and 44 did not giving a total of 314 instead of 320. Is there an explanation for this discrepancy?

10. Table 2 summarizes the stature and nutritional status: Is stature (height) not a parameter for assessing nutrition status? HAZ, WAZ, BMIAZ are all nutritional status indices.

Discussion

1. Growth is measured through nutritional status assessment. Using both the way it is seen in this study is somehow not clear. Increases or decreases in anthropometric parameters (weight, height, BMI) were not studied.

2. Future research should aim to identify and investigate these factors: This better as Future research should aim to investigate and identify these factors.

3. School meals mentioned in this study did not specify if the meals were caregiver provided or government provided. Figure 3 shows that some children in public schools did not consume school meals.

References

1. Ref 18 should be reviewed for journal title and appropriate punctuations.

6. PLOS authors have the option to publish the peer review history of their article (what does this mean? ). If published, this will include your full peer review and any attached files.

**Do you want your identity to be public for this peer review?** For information about this choice, including consent withdrawal, please see our Privacy Policy .

Reviewer #1: **Yes: ** That's Ok

Reviewer #2: **Yes: ** Professor Rufina N. B. Ayogu

---

## [Author Response · Author response to Decision Letter 1]

12 Feb 2025

Response to reviewers

We sincerely appreciate the time and effort of the academic editor and reviewers in evaluating our manuscript.

Reviewer #1

Reviewer comment: The authors have not categorically pointed out the inculsion and exclusion crieteria for the subject sample.

Response: We appreciate the reviewer's observation. The inclusion criteria are implicit in the Materials and methods section, where we describe selecting a random sample of healthy children aged 5–10 years from primary schools and excluding those with chronic health conditions or uncertain dates of birth. However, to address the reviewer’s concern, we have now explicitly listed them in the revised text.

Revised text: 'Inclusion criteria were healthy primary school children aged 5–10 years, while children with chronic health conditions or uncertain birth dates were excluded.'

Reviewer comment: Sample size (506) does not represent the socioeconomic disparities associated with growth and nutritional status of the children at national level hence a larger study is needed to depict if the same situation prevail in the country.

Response: We appreciate the reviewer’s comment. However, our study does not claim to represent the entire country, as is evident from the title and stated aim. The study specifically focuses on primary school children in Wad-Madani City to provide localized insights into socioeconomic disparities in growth and nutritional status. However, as a side note, given the similar circumstances and population structure across many regions in Sudan, we believe there are no strong reasons why these findings may not cautiously apply to other similar settings in the country

Reviewer comment: There have been spelling and grammar mistakes in the text which need to be addressed.

Response: We appreciate the reviewer's feedback. The manuscript has undergone thorough proofreading, and we believe any language issues are minimal. However, we have carefully rechecked the text and made refinements where necessary to ensure correctness.

Reviewer comment: The verbatin while interpreting data has to be avoided as it creats a bit confusion for the readers implying the article should have more clarity.

Response: We thank the reviewer for their comment. However, the concern is not entirely clear. If the issue pertains to the clarity of data interpretation, we have carefully reviewed and refined relevant sections to ensure that our explanations are precise and unambiguous.

Reviewer #2

Reviewer comment: Growth should be expunged in the title as direct growth assessment was not conducted. The growth indices assessed depict nutritional status.

Response: We sincerely appreciate the reviewer's comment. However, we respectfully disagree with the suggestion to remove 'growth' from the title for the following reasons:

• Our study, certainly, assessed growth using standard anthropometric measures:

o We measured height, weight, and BMI and plotted them on established growth charts, which is the standard method for assessing one-point growth status in children.

o This is the same approach used in clinical practice and epidemiological studies to assess child growth (https://www.cdc.gov/growthcharts/background.htm).

• Nutritional status was assessed as a reflection of growth assessment:

o We could not have assessed nutritional status without first assessing growth through these standard anthropometric indices.

o Both assessments were conducted at a single point in time, aligning with widely accepted methodologies (https://www.ncbi.nlm.nih.gov/books/NBK537315).

• Cross-sectional (one-off) growth assessments are an established method in child health research:

o The CDC growth charts, one of the most widely used references, were developed using this exact approach—measuring different children’s growth at different ages rather than tracking growth velocity over time.

o It is common practice in both clinical and public health research to use single-time-point growth assessments to evaluate children's health status (https://pubmed.ncbi.nlm.nih.gov/12043359).

• We used 'growth charts,' not 'nutritional status charts.'

o The charts and Z-score standards we used are called 'growth charts,' not 'nutritional status charts.'

o Removing 'growth' would inaccurately reflect the standard terminology used in growth and nutritional research (https://www.who.int/tools/child-growth-standards/standards).

• Title relevance and study visibility:

o Keeping 'growth' in the title is essential for clarity, accuracy, and search visibility in scientific literature.

o It correctly represents the study’s focus and ensures it is easily identified by relevant readers and researchers.

• Clarification on what we did not assess:

o We did not assess longitudinal growth velocity (growth rate over time), which seems to be the reviewer's concern.

o However, cross-sectional growth assessments remain the primary method for evaluating child growth and nutritional status in epidemiological studies (https://www.cdc.gov/growth-chart-training/hcp/using-growth-charts/who-methodology.html).

Given these points, we respectfully request that the reviewer reconsider this suggestion and allow us to retain 'growth' in the title and relevant sections.

Reviewer comment: Abstract: The method section of the abstract is not detailed enough to show how the objective was achieved. Was comparison of the Z scores between private and public schools the only assessment conducted? The study design, sample size and sampling techniques used are missing. Which data collection methods were employed? How were the data analysed? The title showed that the authors also assessed growth. How was this conducted?

Response: We thank the reviewer for her valuable comments. We have revised the abstract to provide additional methodological details, including the study design, sample size, sampling method, data collection approach, and statistical analyses.

Regarding the reviewer's last point, we have already clarified in our previous response that growth was assessed using standard anthropometric measures (height, weight, and BMI), which were plotted on WHO growth charts. This remains the standard approach for evaluating growth at a single point in time in both clinical and epidemiological settings.

We appreciate the reviewer’s feedback and believe the revised abstract now fully addresses the concerns raised.

Revised abstract:

Objective. To assess the growth and nutritional status of children in primary schools across different socioeconomic groups in Wad-Madani City, Central Sudan, and map it to World Health Organization (WHO) standards; and to investigate a potential association between school meal intake and nutritional status.

Methods. This cross-sectional anthropometric study involved a randomly selected sample of 506 children from 10 primary schools in the city. Height and weight were measured following WHO standards and converted into Z-scores for weight-for-age (WAZ), height-for-age (HAZ), and BMI-for-age (BAZ). We compared the mean Z-scores between children in the private and public school sectors, adjusting for ethnicity and other potential predictors. Statistical analyses included multivariate linear regression to assess predictors of growth and nutritional status, alongside group comparisons using appropriate statistical tests.

Results. Children in public schools had significantly lower BAZ and HAZ levels compared to both WHO standards and private school children. The mean BAZ was -1.0 (SD = 1.23) for public school children and -0.13 (SD = 1.40) for private school children (p = 0.001), with 17.8% (n = 57) of public school children classified as thin (wasted) or severely wasted. The median HAZ was -0.20 (95% CI: -0.34, -0.02) for public school children and 0.19 (95% CI: 0.03, 0.40) for private school children (p < 0.001). Additionally, children in suburban public schools had a significantly lower mean HAZ (-0.46, SD = 11.33) compared to those in urban public schools (p = 0.009). Compared to WHO growth standards, public school children had significantly lower mean WAZ (p < 0.001), HAZ (p = 0.002), and BAZ (p < 0.001). Children who received school meals had significantly higher WAZ (mean difference = 0.619, p = 0.001), HAZ (mean difference = 0.401, p = 0.010), and BAZ (mean difference = 0.588, p = 0.003) across the entire sample. Even within the public-school subgroup, while statistical significance was not reached, all three parameters—WAZ (mean difference = 0.334, p = 0.074), HAZ (mean difference = 0.262, p = 0.123), and BAZ (mean difference = 0.299, p = 0.132)—remained consistently higher among those who received school meals.

Conclusion. Public school children exhibit unfavorable growth and nutritional status, which may be attributed to inadequate nutritional and calorie intake. School meals may improve nutritional outcomes. We propose urgent intervention through the provision of nutritionally adequate school meals.

Reviewer comment: The results are clear but results on growth is missing.

Response: The abstract has been revised to address this. See the above response please.

Reviewer comment: By growth, I expected to see baseline and endline values and mean differences that shows increase or decrease in growth parameters (weight, height, BMI). When these parameters are related to age or any other parameter, it becomes an index of nutritional status.

Response: We thank the reviewer for their comment. However, the expectation of “baseline and endline measurements” reflects growth velocity assessment (growth rate over time), which is a complementary but not primary method for assessing growth in both clinical and epidemiological settings in children. The cross-sectional approach, where anthropometric indices are measured and compared with standardized growth charts according to age and sex, remains the most widely practiced method in child growth assessment.

We also respectfully would like to clarify that in children, all growth parameters must be related to age; otherwise, growth assessment would not be meaningful. No clinically or epidemiologically sound growth assessment is conducted without relating anthropometric measurements to age. This is the basis of all internationally recognized child growth standards (WHO, CDC).

As discussed in our previous response, our study followed this standard cross-sectional methodology, making it an appropriate and widely accepted approach. We hope this clarification addresses the reviewer’s concern.

Reviewer comment: Growth used in the conclusion is inappropriate because growth was not assessed. The authors assessed nutritional status; the growth indicators are not direct assessment of growth pattern. The key words should also be reviewed in line with this statement.

Response: We thank the reviewer for their comment. However, we believe this concern has already been sufficiently addressed in our previous responses. As previously clarified, our study assessed growth using standard anthropometric indicators (weight-for-age, height-for-age, and BMI-for-age), which are internationally recognized methods for assessing child growth at a single point in time. These indicators are referred to as 'growth indicators' in WHO and CDC classifications and are widely used in both clinical and epidemiological research.

The reviewer's concern likely stems from conflating growth assessment with growth velocity tracking over time, which was not the study’s aim. Cross-sectional growth assessment using standardized Z-scores remains the primary and most widely accepted approach for evaluating child growth in epidemiological studies.

Given this, we believe that the term ‘growth’ remains appropriate in the conclusion and keywords.

Reviewer comment: weight for height or body mass index (BMI) for age: these are indices and should be written as compound words.

Response: We thank the reviewer for their observation. Indeed, 'weight-for-height' and 'BMI-for-age' are widely accepted terms in scientific literature and are typically written with hyphens when used as compound adjectives (e.g., weight-for-height Z-score or BMI-for-age reference standard). When used as standalone terms (e.g., weight for height is an indicator of acute malnutrition), hyphens are not required. This convention follows the standard usage in WHO and CDC publications, where 'weight-for-height' and 'BMI-for-age' are written with hyphens when modifying a noun but not necessarily when used independently (please refer to the following content to verify our response: WHO Growth Standards and CDC Growth Chart Training).

Revised text: We have reviewed the manuscript to ensure consistent usage of these terms following standard scientific conventions.

Reviewer comment: We estimated a sample size of 392: This statement is inappropriate. The authors estimated the sample size instead of calculating it based on a prevalence of malnutrition of 15%, a confidence limit of 5%, a cluster effect of 2 and a 95% confidence level. Is there any explanation for this?

Response: We thank the reviewer for their observation. We acknowledge that 'calculated' may be a more precise term than 'estimated' in this context. The minimum sample size was determined using a prevalence of malnutrition of 15%, a confidence limit of 5%, a cluster effect of 2, and a 95% confidence level. To enhance clarity, we have revised the wording in the manuscript to reflect this.

Revised text: With a population of around 90000 primary school children in Wad-Madani, a hypothesized prevalence of malnutrition of 15% [10], a confidence limit of 5% and, a cluster effect of 2, we calculated a minimum required sample size of 392 for a 95% confidence level.

Reviewer comment: The authors estimated a sample size of 392 but distributed 1000 consent forms. What was the essence of estimating (uncalculated) the sample size. It appears that selection of the pupils for the study was dependent on return of consent forms. It is also not clear how they arrived at the 1000 pupils given the forms.

Response: We thank the reviewer for their comment. The calculated sample size of 392 represents the minimum required number of participants to achieve sufficient statistical power. However, we aimed for a higher number to account for potential non-response and missing data. It is common practice to recruit more participants to account for non-response, withdrawals, or missing data.

To maximize recruitment, we distributed 1000 consent forms, anticipating that some parents would not return them. This approach aligns with best practices in epidemiological and school-based research, where oversampling is commonly used to ensure adequate recruitment (https://pmc.ncbi.nlm.nih.gov/articles/PMC5508166/?utm_source=chatgpt.com).

Selection was therefore dependent on the return of consent forms, as is standard in school-based studies requiring parental approval (https://bmjopen.bmj.com/content/13/6/e070277). We have clarified this in the manuscript to ensure transparency.

Revised text: With a population of around 90000 primary school children in Wad-Madani, a hypothesized prevalence of malnutrition of 15% [10], a confidence limit of 5% and, a cluster effect of 2, we calculated a minimum required sample size of 392 for a 95% confidence level.

Reviewer comment: How many children were selected from each of the areas (public/private) schools and on what basis. Same applies to each of the schools.

Response: We thank the reviewer for this comment. The manuscript provides details on the total number of children selected from public and private schools, as well as their distribution across urban and suburban areas. However, data on the exact number of participants from each of the 10 schools was not collected. Selection within each school was not based on proportional allocation to school enrollment size but rather on a random selection of children aged 5–10 years from the list of eligible students in the participating schools.

We have revised the manuscript to explicitly state that the selection within each school was not proportionate to school enrollment size.

Revised text: Selection within each school was not proportionate to school enrollment size; instead, an equal opportunity approach was applied to ensure feas

---

## [Decision Letter · Decision Letter 1]

2 Mar 2025

PONE-D-24-52592R1Socioeconomic Disparities in Growth and Nutritional Status Among Primary School Children in Central Sudan: A Potential Association with School MealsPLOS ONE

Dear Dr. Alhassan,

Thank you for submitting your manuscript to PLOS ONE. After careful consideration, we feel that it has merit but does not fully meet PLOS ONE’s publication criteria as it currently stands. Therefore, we invite you to submit a revised version of the manuscript that addresses the points raised during the review process.

**Dear Alhassan,**You have addressed the comments, but there are still minor comments you need to address.**Best wishes **==============================

We look forward to receiving your revised manuscript.

Kind regards,

Bilal Ahmad Rahimi, M.D., D.T.M.&H., M.C.T.P., Ph.D

Academic Editor

PLOS ONE

Journal Requirements:

Reviewers' comments:

Reviewer's Responses to Questions

**Comments to the Author**

1. If the authors have adequately addressed your comments raised in a previous round of review and you feel that this manuscript is now acceptable for publication, you may indicate that here to bypass the “Comments to the Author” section, enter your conflict of interest statement in the “Confidential to Editor” section, and submit your "Accept" recommendation.

Reviewer #1: All comments have been addressed

Reviewer #2: (No Response)

2. Is the manuscript technically sound, and do the data support the conclusions?

Reviewer #1: Yes

Reviewer #2: Yes

3. Has the statistical analysis been performed appropriately and rigorously? 

Reviewer #1: Yes

Reviewer #2: Yes

4. Have the authors made all data underlying the findings in their manuscript fully available?

Reviewer #1: Yes

Reviewer #2: Yes

5. Is the manuscript presented in an intelligible fashion and written in standard English?

Reviewer #1: Yes

Reviewer #2: Yes

6. Review Comments to the Author

Reviewer #1: The paper seems too lengthy and can be cut short if this does not impact the overall structure and findings of this pieace of research

Reviewer #2: Reviewer’s comment: I have read the authors’ response to the comments on growth and nutrition status. According to UNICEF ‘Weight gain is the most important sign that a child is healthy and is growing and developing well’ This is one of the basis for the suggestion to expunge growth.

https://www.unicef.org/uganda/key-practice-monitoring-growth-and-development-child#:~:text=Weight%20gain%20is%20the%20most,if%20the%20child%20is%20overweight.

Based on this reference, I would like to explain a bit further that when data on weight, height and BMI are obtained using a cross-sectional survey design, the single data are related to the child’s age and sex and compared to child growth standards as explained by the authors. This comparison yields anthropometric indices (weight-for-age and others) which are interpreted to showcase anthropometric status (a component of nutrition status) such as stunting, wasting, thinness and underweight. These can be used to identify children who are growing well. For the term growth to be used in a title of a manuscript, I expected to see weight gain calculated and not necessarily growth velocity/growth rate which are extension of growth assessment. Weight gain or weight loss as shown by UNICEF is the most important indicator of growth status. This is lacking in the manuscript. Besides, nutrition status assessment has 4 components (anthropometric, biochemical, clinical and dietary components). In assessment of nutrition status of individuals, at least 2 components should be used otherwise the specific component assessed (in this case anthropometric status) should be named and not the broad heading (nutritional status) despite the fact that some journals have published them. In the study under discourse, the authors used only anthropometry (one out of four) and therefore should use the term (anthropometric status) instead of nutrition status (for specificity). Again, their use of growth and nutrition status sort of separated growth from nutrition status which is correct because through out the study, they did not show weight status (gain or loss) or linear growth but compared the current anthropometric measurements as obtained cross sectionally with age and sex and interpreted them in line with child growth standards. Comparing anthropometric indices (HAZ, WAZ, BMIAZ, WHZ) with child growth standards would not yield data on growth status (gain or loss) but anthropometric status as normal, retarded (stunting, underweight, thinness and wasting) or above normal (overweight, obesity, too tall). Granted that some published articles may have used this same title, it does not seem to be correct. Based on this, I would suggest a rephrase of the title as Socioeconomic Disparities in Anthropometric Status Among Primary School Children in Central Sudan: A Potential Association with School Meals. This makes the title very specific to what the authors studied. Anthropometric assessment alone cannot showcase nutrition status as explained above. This is because nutrition status is much more than anthropometric status.

A study published in PLOS ONE journal showed growth and nutritional status clearly and there is no doubt about its title. Please see Hamid Namaganda L, Andrews C, Wabwire-Mangen F, Peterson S, Forssberg H, Kakooza-Mwesige A (2023) Nutritional status and growth of children and adolescents with and without cerebral palsy in eastern Uganda: A longitudinal comparative analysis. PLOS Glob Public Health 3(6): e0001241. ttps://doi.org/10.1371/journal.pgph.0001241

The authors may also wish to look at this article for growth assessment: Zoleko-Manego R, Mischlinger J, Dejon-Agobé JC, Basra A, Mackanga JR, Akerey Diop D, et al. (2021) Birth weight, growth, nutritional status and mortality of infants from Lambaréné and Fougamou in Gabon in their first year of life. PLoS ONE 16(2): e0246694. https://doi.org/10.1371/journal.pone.0246694.

Response: We sincerely appreciate the reviewer's comment. However, we respectfully disagree with the suggestion to remove 'growth' from the title for the following reasons:

• Our study, certainly, assessed growth using standard anthropometric measures:

We measured height, weight, and BMI and plotted them on established growth charts, which is the standard method for assessing one-point growth status in children.

Reviewer’s response: When this is done, the outcome is anthropometric status, one aspect of nutrition status assessment.

o This is the same approach used in clinical practice and epidemiological studies to assess child growth (https://www.cdc.gov/growthcharts/background.htm).

Reviewer’s response: Actually, it is to assess anthropometric status an aspect of nutrition status assessment as shown in the result section of this study.

• Nutritional status was assessed as a reflection of growth assessment:

Reviewer’s response: Anthropometric status and not nutrition status (one component out of 4 was assessed)

o We could not have assessed nutritional status without first assessing growth through these standard anthropometric indices.

Reviewer’s response: If you assessed growth, what was the outcome of your assessment? No table showed this.

o Both assessments were conducted at a single point in time, aligning with widely accepted methodologies (https://www.ncbi.nlm.nih.gov/books/NBK537315).

• Cross-sectional (one-off) growth assessments are an established method in child health research:

Reviewer’s response: I would rather say that anthropometric assessments are.

o The CDC growth charts, one of the most widely used references, were developed using this exact approach—measuring different children’s growth at different ages rather than tracking growth velocity over time.

Reviewer’s response: The emphasis is far from growth velocity.

o It is common practice in both clinical and public health research to use single-time-point growth assessments to evaluate children's health status (https://pubmed.ncbi.nlm.nih.gov/12043359).

Reviewer’s response: No doubt about this but it requires further assessment beyond what anthropometry reveals.

• We used 'growth charts,' not 'nutritional status charts.'

Reviewer’s response: There is no thing as nutritional status charts but child growth charts which reveal anthropometric status, a component of nutrition status.

o The charts and Z-score standards we used are called 'growth charts,' not 'nutritional status charts.'

o Removing 'growth' would inaccurately reflect the standard terminology used in growth and nutritional research (https://www.who.int/tools/child-growth-standards/standards).

Reviewer’s response: No, studies should report specifically what they revealed. This study was on anthropometric status of children using anthropometric indices (weight-for-age, height-for-age and BMI-for-age Z scores).

• Title relevance and study visibility:

o Keeping 'growth' in the title is essential for clarity, accuracy, and search visibility in scientific literature.

Reviewer’s response: Only if the authors assessed growth. They assessed anthropometric status using anthropometric indices derived from anthropometric measurements specific to age and sex.

o It correctly represents the study’s focus and ensures it is easily identified by relevant readers and researchers.

• Clarification on what we did not assess:

o We did not assess longitudinal growth velocity (growth rate over time), which seems to be the reviewer's concern.

Reviewer’s response: Of course, no data showed this. The issue is growth status and not growth rate. There is no way this study would show growth rate since it is design is cross sectional.

o However, cross-sectional growth assessments remain the primary method for evaluating child growth and nutritional status in epidemiological studies (https://www.cdc.gov/growth-chart-training/hcp/using-growth-charts/who-methodology.html).

Reviewer’s response: Cross-sectional growth assessments: the correct phrase is: cross sectional anthropometric assessment using WHO child growth standards. These standards relate anthropometric measurements/calculations (weight, height, BMI) to the age and sex of healthy children in the populations to identify those with deficits or excesses. It is not enough to assess nutrition status with only one component. It may be allowed but it is not appropriate to use nutrition status when this is done.

Given these points, we respectfully request that the reviewer reconsider this suggestion and allow us to retain 'growth' in the title and relevant sections.

Reviewer’s response: I am not convinced that the authors assessed growth of the children. They assessed anthropometric status by taking weight and height measurements and calculating BMI in a cross sectional study. They used child growth standards specific to age and sex to evaluate these parameters and arrived at the anthropometric status of the children showcasing their Z scores. The result section showed these clearly.

Reviewer comment: Abstract: The method section of the abstract is not detailed enough to show how the objective was achieved. Was comparison of the Z scores between private and public schools the only assessment conducted? The study design, sample size and sampling techniques used are missing. Which data collection methods were employed? How were the data analysed? The title showed that the authors also assessed growth. How was this conducted?

Response: We thank the reviewer for her valuable comments. We have revised the abstract to provide additional methodological details, including the study design, sample size, sampling method, data collection approach, and statistical analyses.

Regarding the reviewer's last point, we have already clarified in our previous response that growth was assessed using standard anthropometric measures (height, weight, and BMI), which were plotted on WHO growth charts. This remains the standard approach for evaluating growth at a single point in time in both clinical and epidemiological settings.

Reviewer’s response: What do the authors have as outcome of using standard anthropometric measures (height, weight, and BMI), plotted on WHO growth charts? If we take it that growth is assessed using anthropometric indices, what do the authors mean by nutrition status then? Would it not be unnecessary repetition? Anthropometry alone cannot stand for nutrition status because it is only one out of 4 components of nutrition status.

We cannot have growth and at the same time nutrition status in the same title where growth was not assessed to show gain or loss but growth parameters were related to age and sex to arrive at anthropometric indices and thus status. I am not satisfied with the explanation because the specific term assessed is missing in the title. Article titles should be direct, accurate and appropriate, precise and should not be misleading.

Reviewer’s response: How the weight and height values were converted into Z-scores is missing in the manuscript. How did the authors obtain BMI data before converting to Z scores? These details are necessary.

As discussed in our previous response, our study followed this standard cross-sectional methodology, making it an appropriate and widely accepted approach. We hope this clarification addresses the reviewer’s concern.

Reviewer’s response: ‘baseline and endline measurements reflects growth velocity assessment (growth rate over time)’. Not always, it depends on what the authors want to achieve. When the weights are plotted, a graph is produced to reveal weight gain, loss or static weight. The authors assessed anthropometric status of the children using growth parameters of weight, height and BMI. These were compared to sex and age and the indices interpreted to yield anthropometric status which is but one component of nutrition status.

Abstract conclusion

Reviewer’s response: Public school children exhibit unfavorable growth and nutritional status: Does this sentence imply that growth and nutrition status are one word or can be used interchangeably? There is no table or graph showing the growth status to be precise.

Revised text: We have reviewed the manuscript to ensure consistent usage of these terms following standard scientific conventions.

Reviewer’s response: All the anthropometric indices are written as compound words using hyphens, please.

Reviewer’s response: The use of the word around in ‘around 90000 primary school children’ showed approximation. Is it that the authors could not assess school registers to arrive at the exact population?

Revised text: A simple random sample of children aged 5–10 years was selected from each school using a list of enrollees in grades matching the eligible age groups. A simple manual selection method was applied, where names of all listed students were drawn randomly by the investigator. Those selected were checked for eligibility, and if found ineligible, replacements were drawn using the same random selection process."

Reviewer’s response: This is fine but it is obvious the authors do not want to disclose the number selected per school. It would have been good to see how the 1000 consent forms were distributed among the 10 schools.

Reviewer’s response: I would want to believe that the entire 10 schools had population that represented them well.

Reviewer’s response: Table 1: Please indicate that the ages are in years.

Reviewer’s response: Table 2: The reason for using Nutritional Status and Growth Z-scores in the same title is not clear.

Reviewer’s response: Title of Table 2 should be adjusted to: Mean Growth Z-scores and anthropometric status of the children.

7. PLOS authors have the option to publish the peer review history of their article (what does this mean? ). If published, this will include your full peer review and any attached files.

**Do you want your identity to be public for this peer review?** For information about this choice, including consent withdrawal, please see our Privacy Policy .

Reviewer #1: No

Reviewer #2: **Yes: ** Rufina Ayogu

---

## [Author Response · Author response to Decision Letter 2]

3 Mar 2025

Responses to reviewers’ comments

Reviewer #1

Comment: The paper seems too lengthy and can be cut short if this does not impact the overall structure and findings of this pieace of research

Response: We thank the reviewer for this suggestion. However, as no specific sections were specified for reduction, and the manuscript remains within the journal’s word limit, we believe maintaining the current length is necessary to preserve the clarity and completeness of our report.

Reviewer #2

Comment (all addressing same point): I have read the authors’ response to the comments on growth and nutrition status. According to UNICEF ‘Weight gain is the most important sign that a child is healthy and is growing and developing well’ This is one of the basis for the suggestion to expunge growth.

https://www.unicef.org/uganda/key-practice-monitoring-growth-and-development-child#:~:text=Weight%20gain%20is%20the%20most,if%20the%20child%20is%20overweight.

Based on this reference, I would like to explain a bit further that when data on weight, height and BMI are obtained using a cross-sectional survey design, the single data are related to the child’s age and sex and compared to child growth standards as explained by the authors. This comparison yields anthropometric indices (weight-for-age and others) which are interpreted to showcase anthropometric status (a component of nutrition status) such as stunting, wasting, thinness and underweight. These can be used to identify children who are growing well. For the term growth to be used in a title of a manuscript, I expected to see weight gain calculated and not necessarily growth velocity/growth rate which are extension of growth assessment. Weight gain or weight loss as shown by UNICEF is the most important indicator of growth status. This is lacking in the manuscript. Besides, nutrition status assessment has 4 components (anthropometric, biochemical, clinical and dietary components). In assessment of nutrition status of individuals, at least 2 components should be used otherwise the specific component assessed (in this case anthropometric status) should be named and not the broad heading (nutritional status) despite the fact that some journals have published them. In the study under discourse, the authors used only anthropometry (one out of four) and therefore should use the term (anthropometric status) instead of nutrition status (for specificity). Again, their use of growth and nutrition status sort of separated growth from nutrition status which is correct because through out the study, they did not show weight status (gain or loss) or linear growth but compared the current anthropometric measurements as obtained cross sectionally with age and sex and interpreted them in line with child growth standards. Comparing anthropometric indices (HAZ, WAZ, BMIAZ, WHZ) with child growth standards would not yield data on growth status (gain or loss) but anthropometric status as normal, retarded (stunting, underweight, thinness and wasting) or above normal (overweight, obesity, too tall). Granted that some published articles may have used this same title, it does not seem to be correct. Based on this, I would suggest a rephrase of the title as Socioeconomic Disparities in Anthropometric Status Among Primary School Children in Central Sudan: A Potential Association with School Meals. This makes the title very specific to what the authors studied. Anthropometric assessment alone cannot showcase nutrition status as explained above. This is because nutrition status is much more than anthropometric status.

A study published in PLOS ONE journal showed growth and nutritional status clearly and there is no doubt about its title. Please see Hamid Namaganda L, Andrews C, Wabwire-Mangen F, Peterson S, Forssberg H, Kakooza-Mwesige A (2023) Nutritional status and growth of children and adolescents with and without cerebral palsy in eastern Uganda: A longitudinal comparative analysis. PLOS Glob Public Health 3(6): e0001241. ttps://doi.org/10.1371/journal.pgph.0001241

The authors may also wish to look at this article for growth assessment: Zoleko-Manego R, Mischlinger J, Dejon-Agobé JC, Basra A, Mackanga JR, Akerey Diop D, et al. (2021) Birth weight, growth, nutritional status and mortality of infants from Lambaréné and Fougamou in Gabon in their first year of life. PLoS ONE 16(2): e0246694. https://doi.org/10.1371/journal.pone.0246694.

When this is done, the outcome is anthropometric status, one aspect of nutrition status assessment.

Actually, it is to assess anthropometric status an aspect of nutrition status assessment as shown in the result section of this study.

Anthropometric status and not nutrition status (one component out of 4 was assessed).

If you assessed growth, what was the outcome of your assessment? No table showed this.

I would rather say that anthropometric assessments are.

The emphasis is far from growth velocity.

No doubt about this but it requires further assessment beyond what anthropometry reveals.

There is no thing as nutritional status charts but child growth charts which reveal anthropometric status, a component of nutrition status.

No, studies should report specifically what they revealed. This study was on anthropometric status of children using anthropometric indices (weight-for-age, height-for-age and BMI-for-age Z scores).

Only if the authors assessed growth. They assessed anthropometric status using anthropometric indices derived from anthropometric measurements specific to age and sex.

Of course, no data showed this. The issue is growth status and not growth rate. There is no way this study would show growth rate since it is design is cross sectional.

Cross-sectional growth assessments: the correct phrase is: cross sectional anthropometric assessment using WHO child growth standards. These standards relate anthropometric measurements/calculations (weight, height, BMI) to the age and sex of healthy children in the populations to identify those with deficits or excesses. It is not enough to assess nutrition status with only one component. It may be allowed but it is not appropriate to use nutrition status when this is done.

I am not convinced that the authors assessed growth of the children. They assessed anthropometric status by taking weight and height measurements and calculating BMI in a cross sectional study. They used child growth standards specific to age and sex to evaluate these parameters and arrived at the anthropometric status of the children showcasing their Z scores. The result section showed these clearly.

What do the authors have as outcome of using standard anthropometric measures (height, weight, and BMI), plotted on WHO growth charts? If we take it that growth is assessed using anthropometric indices, what do the authors mean by nutrition status then? Would it not be unnecessary repetition? Anthropometry alone cannot stand for nutrition status because it is only one out of 4 components of nutrition status.

We cannot have growth and at the same time nutrition status in the same title where growth was not assessed to show gain or loss but growth parameters were related to age and sex to arrive at anthropometric indices and thus status. I am not satisfied with the explanation because the specific term assessed is missing in the title. Article titles should be direct, accurate and appropriate, precise and should not be misleading.

‘Baseline and endline measurements reflects growth velocity assessment (growth rate over time)’. Not always, it depends on what the authors want to achieve. When the weights are plotted, a graph is produced to reveal weight gain, loss or static weight. The authors assessed anthropometric status of the children using growth parameters of weight, height and BMI. These were compared to sex and age and the indices interpreted to yield anthropometric status which is but one component of nutrition status.

Public school children exhibit unfavorable growth and nutritional status: Does this sentence imply that growth and nutrition status are one word or can be used interchangeably? There is no table or graph showing the growth status to be precise.

Response: We appreciate your comment. The title has been changed as per the reviewer’s comment.

Comment: The use of the word around in ‘around 90000 primary school children’ showed approximation. Is it that the authors could not assess school registers to arrive at the exact population?

Response: We thank the reviewer for this comment. The school registries do not contain precise records of student enrollment. Therefore, we relied on the official estimates provided by the State Ministry of Education, which represent the best available data for the primary school population.

Comment: This is fine but it is obvious the authors do not want to disclose the number selected per school. It would have been good to see how the 1000 consent forms were distributed among the 10 schools.

Response: We thank the reviewer for this comment. Unfortunately, we do not have a precise record of the distribution of consent forms across the 10 schools. However, we believe this does not significantly impact the methodological rigor of the selection process.

Comment: I would want to believe that the entire 10 schools had population that represented them well.

Response: We thank the reviewer for this comment. As no specific change or recommendation was provided, we believe no revisions to the manuscript are necessary at this stage.

Comment: Table 1. Please indicate that the ages are in years.

Response: Thank you, this has been indicated.

Comment: Table 2: The reason for using Nutritional Status and Growth Z-scores in the same title is not clear.

Response: We appreciate the reviewer’s feedback. The title was changed to “Mean Growth Z-scores and anthropometric status of the children”.

---

## [Editor Report · Decision Letter 2]

4 Mar 2025

Socioeconomic Disparities in Anthropometric Status Among Primary School Children: A Potential Association with School Meals

PONE-D-24-52592R2

Dear Dr. Alhassan,

We’re pleased to inform you that your manuscript has been judged scientifically suitable for publication and will be formally accepted for publication once it meets all outstanding technical requirements.

Kind regards,

Bilal Ahmad Rahimi, M.D., D.T.M.&H., M.C.T.P., Ph.D

Academic Editor

PLOS ONE
---

## [Editor Report · Acceptance letter]

PONE-D-24-52592R2

PLOS ONE

Dear Dr. Alhassan,

I'm pleased to inform you that your manuscript has been deemed suitable for publication in PLOS ONE. Congratulations! Your manuscript is now being handed over to our production team.

Kind regards,

on behalf of

Professor Bilal Ahmad Rahimi

Academic Editor

PLOS ONE